# Efficacy and Safety of Tinzaparin Thromboprophylaxis in Lung Cancer Patients with High Thromboembolic Risk: A Prospective, Observational, Single-Center Cohort Study

**DOI:** 10.3390/cancers16071442

**Published:** 2024-04-08

**Authors:** Marousa Kouvela, Maria Effrosyni Livanou, Dimitra T. Stefanou, Ioannis A. Vathiotis, Fotini Sarropoulou, Maria Grammoustianou, Evangelos Dimakakos, Nikolaos Syrigos

**Affiliations:** 1Oncology Unit, Third Department of Internal Medicine, Sotiria General Hospital for Chest Diseases, National and Kapodistrian University of Athens, 11527 Athens, Greece; mirsinilivanou20@gmail.com (M.E.L.); johnvathiotis1@gmail.com (I.A.V.); sphotini@hotmail.com (F.S.); mgrammoustianou.trials@gmail.com (M.G.); edimakakos@yahoo.gr (E.D.); nksyrigos@gmail.com (N.S.); 2First Department of Internal Medicine, Laikon General Hospital, School of Medicine, National Kapodistrian University of Athens, 11527 Athens, Greece; dimitroulastef@hotmail.com

**Keywords:** lung cancer, thromboprophylaxis, tinzaparin, venous thromboembolic disease

## Abstract

**Simple Summary:**

Cancer is implicated in multiple pathways that increase thrombogenicity, and lung cancer patients have a 20% higher risk of venous thromboembolism in comparison to the general population. Venous thromboembolic disease (VTE) in cancer patients, which includes deep vein thrombosis (DVT) and pulmonary embolism (PE), can lead to the delay of cancer treatment and, thus, result in increased mortality, morbidity, and burden on healthcare resources. Factors contributing to thrombotic burden are related to cancer, patients, treatment, and laboratory findings. Thromboprophylaxis during active lung cancer treatment with adequate anticoagulation might improve outcomes. Thromboprophylaxis with low molecular weight heparin (LMWH) is the standard of care, but due to the vast heterogeneity of lung cancer patients, there is no consensus on the optimal dose and duration of the treatment.

**Abstract:**

Background: The aim of this study was to record and assess the efficacy and safety ofthromboprophylaxis with an intermediate dose of Tinzaparin in lung cancer patients with high thrombotic risk. Methods: This was a non-interventional, single-arm, prospective cohort study of lung cancer patients who received thromboprophylaxis with Tinzaparin 10.000 Anti-Xa IU in 0.5 mL, OD, used in current clinical practice. Enrolled ambulatory patients signed informed consent. Anti-Xa levels were tested. Results: In total, 140 patients were included in the study, of which 81.4% were males. The histology of the tumor was mainly adenocarcinoma. Lung cancer patients with high thrombotic risk based on tumor, patient, treatment, and laboratory-related factors were enrolled. Only one patient experienced a thrombotic event (0.7%), and 10 patients had bleeding events (7.1%), including only one major event. Anti-Xa levels measured at 10 days and 3 months did not differ significantly between patients who developed hemorrhagic events and those who did not (*p* = 0.26 and *p* = 0.32, respectively). Conclusion: Thromboprophylaxis with an intermediate Tinzaparin dose in high thrombotic-risk lung cancer patients is a safe and effective choice for the prevention of VTE.

## 1. Introduction

Venous thromboembolic disease (VTE), which includes deep vein thrombosis (DVT) and pulmonary embolism (PE), is a common complication of malignancy. The relationship between cancer and thrombosis was first identified more than a century ago by Trousseau, and it is now estimated that up to 20% of patients with cancer develop VTE [1,2,3]. VTE in cancer patients is associated with a multitude of adverse outcomes, including increased morbidity and mortality, the postponement of therapy, the need for long-term anticoagulation with the potential for bleeding complications, and high rates of recurrent VTE [4,5]. In addition, it leads to a significant increase in healthcare resource utilization. In one study of cancer patients, the adjusted mean all-cause additional healthcare costs of VTE were USD 30,538 per patient [6]. The risk varies with the type of cancer, as well as the stage of the disease. Other risk factors such as age, gender, bed rest, venous catheters, surgery, chemotherapy with or without adjuvant hormone therapy, radiation therapy, and infections also increase the risk of thrombosis in cancer patients. Cancer-associated thrombosis involves a complex interplay between direct cancer cell-mediated pathways and indirect host cell-mediated mechanisms. Direct mechanisms include the expression or secretion by cancer cells of factors implicated in both primary and secondary hemostasis pathways. Some of the key mediators in the activation of the coagulation cascade include tissue factor (TF), Phosphatidyl serine (PS), the Cancer procoagulant (CP), and cancer microparticles. Furthermore, cancer cells promote platelet activation and aggregation through the expression of Podoplanin (PDPN) and the secretion of platelet agonists such as ADP and thrombin Plasminogen activation inhibitor-1 (PAI-1). Indirect mechanisms include The immune-mediated secretion of cytokines, which promote platelet activation and endothelial inflammation, as well as neutrophil extracellular traps (NETs), which serve as scaffolds to entrap platelets and red blood cells and further promote platelet activation [7].

Lung cancer belongs to the group of malignancies with the highest incidence rates of VTE [2,8,9,10]. Retrospective studies associate adenocarcinoma histology with The increased risk of VTE [11,12]. Blom et al. examined the thrombotic risk in 537 patients with NSCLC and found that it was 20 times higher compared to the general population [standardized morbidity ratio: 20.0 (95% Confidence Interval (CI), 14.6–27.4)]. The risk was three times higher in patients with adenocarcinoma compared to squamous cell carcinoma (incidence 66.7 vs. 21.2 per 1000 person-years) [13]. In recent years, VTE in lung cancer patients has received increasing attention. In the retrospective analysis of a lung cancer cohort of 6732 patients (control group 17,284 patients), VTE occurred in 13.9% of patients in the lung cancer cohort and 1.4% in the control cohort [14]. Among the prothrombotic mechanisms of cancer described above, increased levels of leukocytes, NETs, tissue factor-positive (TF+) microvesicles (MVs), and endothelial cell activation exerted an important role in lung cancer patients [15,16].

Systemic anticancer treatment further increases the risk for VTE in these patients. Specifically, chemotherapy is estimated to account for a 4-7-fold increase [17]. Among lung cancer patients receiving chemotherapy, the majority of VTE events occur within 6 months of starting chemotherapy [18]. In the CANTARISK study, out of 1980 patients with lung cancer treated in the pre-immunotherapy era, the 6-month incidence of VTE was 6.1% [19]. Immunotherapy, which has become the standard of care for advanced disease, has also been associated with an increased risk of VTE. In a study of 1686 patients with cancer who received immunotherapy, the 6-month incidence of VTE was 7.1% [20]. In another report of 522 immunotherapy-treated patients with lung cancer, the incidence of VTE occurred at 30.3% [21]. Furthermore, VTE in immunotherapy patients was associated with worse survival, but this association was not statistically significant when adjusting for age and metastasis [HR = 1.215, (95% CI 0.94 to 1.55) *p*-value = 0.121] [21]. The mechanism underlying the increased likelihood of venous thromboembolic events among immunotherapy-treated patients is not clear. Two of the mechanisms proposed by Goel et al. are cancer-mediated T-cell activation, leading to subsequent monocyte activation and tissue factor release, and immune-mediated vasculitis, resulting in endothelial damage [22].

In several clinical scenarios of high thromboembolic risk, low molecular weight heparins (LMWH) are safe and effective at preventing VTE [23,24,25,26,27,28]. Current ESMO and ASCO guidelines suggest considering thromboprophylaxis with either Direct Oral Anticoagulants (DOACs) or with low molecular weight heparin (LMWH) in ambulatory high-risk patients [29]. In clinical practice, the main factors affecting physicians’ decisions to use thromboprophylaxis in cancer patients are the Eastern European Cooperative Oncology (ECOG) group score, cancer type, advanced stage, malignancy, chemotherapy, co-morbidities and history of thrombosis [30].

The optimal dosage of LMWH treatment for thromboprophylaxis is not well established. There are data from studies using higher than usual doses for prophylaxis or using full therapeutic doses. For Tinzaparin, a prophylactic dose is considered a dose of 4500 anti-Factor Xa IU/mL, the intermediate dose is 10,000 anti-Factor Xa IU/mL, and the recommended dose is 175 anti-Xa IU/kg of body weight, administered subcutaneously once daily. Intermediate doses were used in the study by Pelzer in 2015, and therapeutic doses in the study by Maraveyas in 2012. Two studies initially administered a therapeutic dose of LMWH followed by intermediate doses (Klerk 2005, van Doormaal 2011) [31,32,33,34]. Regarding the optimal duration of treatment, a systematic review and meta-analysis published in 2020 showed that the extension of treatment beyond six months did not lead to superior efficacy but increased toxicity [27].

Based on the above evidence, we performed a prospective study of Tinzaparin thromboprophylaxis for lung cancer patients at high risk for thrombosis to obtain and evaluate data on efficacy, safety, and patient compliance. The primary objective of this study was to assess the frequency of all venous thromboembolism (VTE) events during the six-month treatment period to assess the frequency of major and minor bleeding events. Secondary objectives were to assess patient compliance and to assess the frequency of bleeding events in relation to anti-Xa levels and to compare the frequency of events between patients receiving and not receiving immune checkpoint inhibitors (ICIs).

## 2. Materials and Methods

This was a non-interventional, single-arm, prospective cohort study of consecutive lung cancer patients who received thromboprophylaxis with Tinzaparin, conducted at the Oncology Unit of the Third Department of Internal Medicine, “Sotiria” General Hospital for Chest Diseases between June 2021 and June 2022. The study protocol was approved by the Local Ethics Committee Review Board. All patients included in the study provided written informed consent. Ambulatory patients with histologically or cytologically confirmed lung cancer who fulfilled the following additional criteria were eligible for study inclusion: patients who were either receiving or were expecting to receive thromboprophylaxis based on current clinical practice; aged over 18 years; and life-expectancy over 6 months at the time of study inclusion. Patients were evaluated for study inclusion at the time of 1st or 2nd lung cancer treatment administration. Thromboprophylaxis was administered by the prescribing doctor based on the common local clinical practice at the time in patients with at least two of the following risk factors:Time since cancer diagnosis < 6 months.Metastatic cancer or high burden of disease (stage ≥ ΙΙΙB).Platinum-based chemotherapy.Antiangiogenesis therapy.Immunotherapy.Platelets > 350.000/μL.Hemoglobin < 10 g/dL.White blood cell count > 11,000/μL.Obesity (BMI > 35).Blood transfusion or use of hematopoietic factors.Recent hospitalization.Reduced mobility.History of deep venous thrombosis (DVT).Congenital thrombophilia (i.e., Factor V Leiden thrombophilia, prothrombin G20210A, Antithrombin III insufficiency, Protein C or protein S insufficiency, etc.).At least two of the following vascular risk factors: a history of peripheral arterial disease (PAN), cerebrovascular accident (CVA), coronary artery disease (CAD), hypertension, dyslipidemia, or diabetes mellitus.Atrial fibrillation.

Patients with at least 2 of these factors received an intermediate dose of Tinzaparin (10.000 anti-XaIU OD) daily during cancer treatment and a maximum of 6 months. The dose and duration of therapy were selected based on previous research showing no superior efficacy from increased dose or duration [24]. The following information was collected for each patient: the date of thromboprophylaxis onset, histology, anticancer treatment (type, agents, and line), risk factors for thrombotic events as described above, thromboembolic events (type and date), hemorrhagic events (date and type), date of tinzaparin discontinuation, date of disease progression and date of death (when applicable). Anti-Xa levels were measured 10 days and 3 months after the onset of Tinzaparin prophylaxis. Additionally, patients with adenocarcinoma were assessed for mutations in 58 genes through Next Generation Sequencing (NGS), as per standard local clinical practice. Specific investigations to detect DVT and PE were performed as requested based on the clinical suspicion of VTE on subsequent patient evaluations. Cases of incidentally diagnosed VTE during routine follow-up imaging were included.

Major bleeding was defined per the ISTH criteria [35] as follows:Fatal bleeding;Symptomatic bleeding in a critical area or organ, such as intracranial, intraspinal, intraocular, retroperitoneal, intraarticular or pericardial, or intramuscular with compartment syndrome;Bleeding causes a fall in hemoglobin level of 2 g/dL or more, leading to the transfusion of two or more units of whole blood or red blood cells.

Statistical analysis was performed using R version 4.2.1. Patient demographics and disease characteristics were analyzed using descriptive statistics. The Pearson chi-squared test was used to assess differences in categorical variables. The Wilcoxon Rank Sum test was used for continuous variables. To account for the lack of a comparative group in the study, as an indirect measure of efficacy comparison, the expected rates of VTE and hemorrhagic events with thromboprophylaxis administered were calculated based on the reports of VTE incidence in the CANTARISK study, which was a global, real-world study of 1980 patients with lung cancer followed-up for 6 months [19]. The risk ratio (RR) was calculated as the ratio of the probability of a thrombotic event in the Tinzaparin group to the expected probability if prophylaxis was not administered, using the Cochran–Mantel–Haenszel test. All hypothesis testing was conducted at a two-sided significance level of α = 0.05.

## 3. Results

### 3.1. Population

In total, 140 patients were included in the analysis. The follow-up period was 6 months. The demographic characteristics of patients are summarized in Table 1. The majority of included patients were males (81.4%), and the median age was 66 (range 46–92). The most prevalent histologic subtype was adenocarcinoma (*N* = 73, 52%), followed by squamous cell carcinoma (*N* = 38, 27%), small-cell lung cancer (SCLC) (*N* = 20, 14%) and others (*N* = 9, 6.4%). The majority of patients (*N* = 90, 64%) were on first-line anticancer treatment at study inclusion. All patients but one had at least two risk factors for VTE at the baseline (as per protocol), with a median number of four (range 1–8). The most common risk factors at the baseline were the high burden of disease (stage > IIIB), cancer diagnosis within 6 months before treatment onset, platinum-based chemotherapy, and immunotherapy. Sixteen (11%) patients had a history of DVT (Table 2).

### 3.2. Tinzaparin Treatment

In total, 135 (96%) patients received intermediate doses of Tinzaparin as per the protocol, while 5 (3.6%) received therapeutic doses. The median time to Tinzaparin treatment discontinuation was 157 days [interquartile range (IQR), 85, 183]. Regarding the reasons for discontinuation, 65 (46.4%) patients completed 6 months of Tinzaparin prophylaxis, 17 (12%) completed the anticancer treatment and hence were no longer eligible for thromboprophylaxis, 24 (17%) died, 5 (3.6%) changed lines of treatment following PD and stopped fulfilling the minimum criteria for thromboprophylaxis, 18 (12.7%) had an adverse event (AE) that prompted the physician to discontinue treatment, 8 (5.7%) discontinued because of the physician’s decision for a reason other than AE, and lastly, 3 (2.1%) patients discontinued of their own will (Table 3).

### 3.3. Efficacy and Safety of Tinzaparin Thromboprophylaxis

Out of 140 patients, only 1 (0.7%) developed radiologically confirmed DVT. This patient subsequently developed PE despite switching to a therapeutic dose of Tinzaparin following the DVT diagnosis.

Based on historical data from the CANTARISK study, the expected incidence of VTE in a similar population not treated with Tinzaparin would be 6.1% or 8 in 140 patients. Using the Mantel–Haenszel method, the relative risk (RR) for thromboembolism between these groups was 0.13 (95% CI, 0.02; 0.99, *p* = 0.048).

By the end of follow-up, 10 patients (7.1%) had developed hemorrhagic events, 9 had minor events, while 1 patient had a major event; specifically, they experienced the hemorrhagic turnover of brain metastasis. The total number of minor hemorrhagic events was 12: 9 episodes of hemoptysis, 2 episodes of rhinorrhagia, and 1 episode of bloody stools. Anti-Xa levels measured at 10 days and at 3 months did not differ significantly between patients who developed hemorrhagic events and those who did not (*p* = 0.26 and *p* = 0.32, respectively) (Table 4).

Thromboembolic and treatment-adverse events are summarized in Table 4. A swimmer’s plot demonstrating the timing of all events (thromboembolic, hemorrhagic, and other adverse events) is presented in Figure 1. Treatment-adverse effects included acute renal failure, allergic reaction, hematologic toxicity, anemia, and thrombocytopenia. The respective frequencies were 1 (0.7%), 2 (1.4%), 2 (1.4%), 1 (0.7%), and 3 (2.1%). 

Regarding patients with a history of VTE prior to the onset of Tinzaparin thromboprophylaxis, no patients had thromboembolic or hemorrhagic events in the 6 months of observation.

### 3.4. Subgroup Analysis—ICIs vs. non-ICIs

Of the 140 patients, 62 (44.3%) received immunotherapy (IO). In the subgroup comparison of IO vs. non-IO-treated patients, patients who received IO had a lower rate of AE’s [4 (6.5%) vs. 14 (18%), *p* = 0.06, q = 0.3]. Furthermore, patients on IO, compared to the rest of the study population, were less likely to have a history of DVT at the baseline (4.8% vs. 17%). Still, the only patient who developed DVT during the six months of observation was in the IO group.

## 4. Discussion

In our cohort, the incidence of thromboembolic events was 0.7%. To make up for the lack of a comparative arm, we employed historical data to estimate the expected incidence of thromboembolic events in a cohort of an equivalent size, assuming that they did not receive thromboprophylaxis [16]. In this indirect comparison, we estimated a lower risk for venous thromboembolism (VTE) in patients treated with Tinzaparin (RR = 0.13, *p* = 0.048); however, the 95% CI was wide (0.02; 0.99), suggesting imprecision. Furthermore, we should acknowledge the inherent limitations associated with this methodology. Firstly, the historical cohort comprised a global population, potentially differing from our single-institution study group. Secondly, the historical data were applied to patients treated during the years 2011–2012. Since then, significant changes have occurred in standard treatment practices, with the advent of immunotherapy being the most notable advancement. Lastly, there have been observed temporal changes in thrombotic risk among cancer patients, with an increase in VTE incidence among cancer patients in recent years, and the introduction of immunotherapy may partly account for this increase. These changes in VTE incidence and current treatment practice may have led to an underestimation of the relative efficacy of Tinzaparin thromboprophylaxis. The limitations mentioned above emphasize the need for cautious interpretation and larger-scale research.

Regarding the safety of Tinzaparin thromboprophylaxis, the rate of hemorrhagic events was 7.1%, which was on the lower end of what has been reported in similar research with a focus on Tinzaparin use in cancer patients for the long-term treatment of VTE [3]. Anti-Xa levels did not differ significantly between patients with hemorrhagic events, which is in accordance with previously published research and endorses how monitoring in patients receiving LMWH prophylaxis is not necessary. Furthermore, they did not differ at 10 days and 3 months, suggesting that the daily intermediate dose of Tinzaparin suffices to achieve stable concentrations. This aligns with recent data suggesting that the use of intermediate doses of Tinzaparin may be more effective than prophylactic doses without safety concerns [36].

The anti-Xa levels at day 10 and 3 months in the IO group were numerically lower than the non-IO treated group [Median (IQR); 0.46 (0.30, 0.59) vs. 0.52 (0.33, 0.65) at day 10 and 0.37 (0.22, 0.60) vs. 0.52 (0.32, 0.75) at 3 months]. Although the study was not adequately powered to detect statistical differences in the rate of events between these subgroups, given the probable association of higher anti-Xa levels with a decrease in the likelihood of thrombotic events and the increase in the likelihood of hemorrhagic events [23], it is plausible that a difference in anti-Xa levels might have contributed to the numerically higher rate of thrombotic events and lower rate of hemorrhagic events in the IO group. More research is needed to elucidate how ICIs affect the anti-Xa levels to assess whether dose adjustment is needed in IO-treated patients.

Finally, although prophylactic treatment required daily injections for 6 months, patient compliance was high, with only three (2.1%) patients opting for early thromboprophylaxis discontinuation.

It is important to acknowledge the limitations of this study. Firstly, since it is non-comparative, it cannot lead to conclusions about the superiority or inferiority of prophylactic Tinzaparin treatment compared to no treatment. Furthermore, the small number of events limits the ability for hypothesis testing and exploratory biomarker analysis. Also, as this was a non-interventional study, anti-Xa could not be measured by our laboratory as we could not perform any interventions on the patients. This is why the measurement was performed externally, and the patients presented an examination on their next scheduled treatment. Finally, the single-center design limits the generalizability of results.

## 5. Conclusions

During the six months of observation, the frequency of VTE was lower than expected based on historical data had prophylaxis not been administered. Furthermore, the frequency of bleeding events was low, with only one major event. This indicates that thromboprophylaxis with intermediate doses of Tinzaparin in patients with lung cancer may be an effective and safe strategy. However, larger-scale comparative research is needed to establish the efficacy, safety, and impact on the survival of outpatient Tinzaparin thromboprophylaxis for patients with cancer receiving systemic treatment, especially for immunotherapy-treated patients where data are more immature.

## Figures and Tables

**Figure 1 cancers-16-01442-f001:**
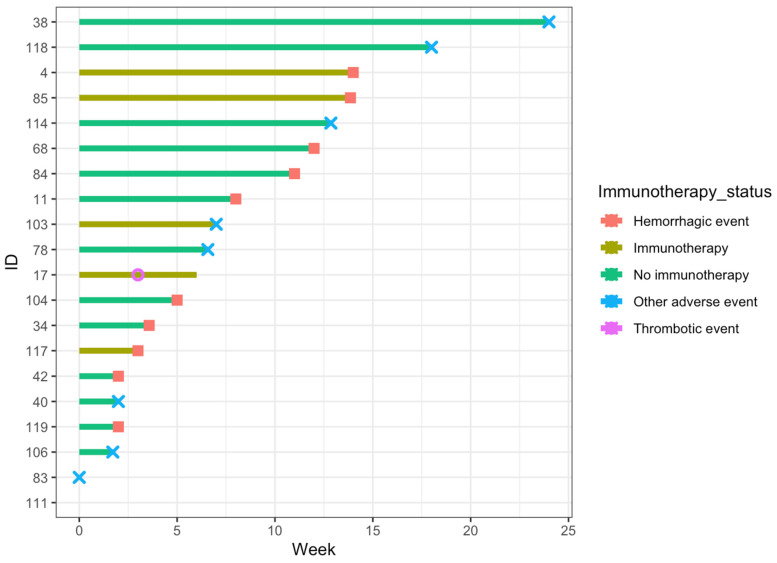
Swimmers plot for patients who developed any event during Tinzaparin thromboprophylaxis. The *x*-axis represents the time in weeks since the onset of Tinzaparin thromboprophylaxis, and the *y*-axis represents the unique identification number assigned to each patient.

**Table 1 cancers-16-01442-t001:** Demographic characteristics of lung cancer patients with high thrombotic risk who were enrolled in the study.

Age, Median (IQR)	66 (60, 74)
Sex, *N* (%)	
Male	114 (81.4)
Female	26 (18.6)
Histology, *N* (%)	
Adenocarcinoma	73 (52)
Squamous	38 (27)
SCLC	20 (14)
Other	9 (6.4)
Line of treatment, *N* (%)	
1	90 (64)
2	24 (17)
3	14 (10)
4	4 (2.9)
Other	8 (5.7)
Immunotherapy (Yes/No), *N* (%)	62 (44)/78 (66)
Line of treatment change during the study	9 (6.4)
Number of VTE risk factors, median (range)	4 (1–8)
1, *N* (%)	1 (0.7)
2, *N* (%)	16 (11)
3, *N* (%)	44 (31)
4, *N* (%)	45 (32)
5, *N* (%)	24 (17)
>5, *N* (%)	10 (7.1)

**Table 2 cancers-16-01442-t002:** Break-down of risk factors for venous thromboembolism at baseline.

Risk Factor	*N* of Patients (%)
Cancer-related	
Stage ≥ IIIB	116 (83)
Diagnosis within the last 6 months	128 (91)
Treatment-related	
Platinum-based chemotherapy	95 (68)
Antiangiogenesis therapy	4 (2.9)
Immunotherapy	95 (68)
Anemia requiring erythropoietin or transfusion	2 (1.4)
Recent hospitalization	8 (5.7)
Laboratory	
Platelets > 350,000/μL	37 (26)
Hemoglobin < 10 g/dL	10 (7.1)
White blood cell count > 11,000/μL	25 (18)
Patient-related	
BMI > 30 kg/m^2^	3 (2.1)
Reduced mobility	14 (10)
History of deep vein thrombosis	16 (11)
Thrombophilia	0 (0)
Vascular risk factors	44 (31)
Atrial fibrillation	7 (5)

**Table 3 cancers-16-01442-t003:** Characteristics of tinzaparin treatment for lung cancer patients with high thrombotic risk enrolled in the study.

Tinzaparin dose, *N* (%)	
Full	5 (3.6)
Intermediate	135 (96)
AntiXa factor levels, median, (IQR)	
On Day 10 (*N* = 120)	0.48 (0.32, 0.61)
At 3 months (*N* = 66)	0.49 (0.28, 0.65)
Treatment duration (Days), median (IQR)	152 (80, 183)
Tinzaparin discontinuation reason, *N* (%)	
Completed 6 months of Tinzaparin prophylaxis	65 (46.4)
Completed anticancer treatment	17 (12.1)
Death	24 (17.1)
Adverse event	18 (12.7)
Line of therapy change *	5 (3.6)
Physician’s decision	8 (5.7)
Patient desire	3 (2.1)

* Only patients who stopped fulfilling the criteria for Thromboprophylaxis.

**Table 4 cancers-16-01442-t004:** Adverse events that occurred during the treatment period with tinzaparin.

Thromboembolic events, *N* of patients (%)	2 (1.4)
DVT, *N* of events (%)	2 (1.4)
PE, *N* of events (%)	1 (0.7)
Hemorrhagic events, *N* of patients (%)	10 (7.1)
Hemoptysis, *N* of events (%)	9 (6.4)
Hemorrhagic turnover of brain metastasis, *N* of events (%)	1 (0.7)
Rhinorrhagia, *N* of events (%)	2 (1.4)
Bloody stool, *N* of events (%)	1 (0.7)
Other adverse events, *N* (%)	
Acute renal failure, *N* of events (%)	1 (0.7)
Allergic reaction, *N* of events (%)	2 (1.4)
Anemia, *N* of events (%)	1 (0.7)
Hematologic toxicity, *N* of events (%)	2 (1.4)
Thrombocytopenia, *N* of events (%)	3 (2.1)

## Data Availability

The data are contained within the article.

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
