# Peer review of "Efficacy and Safety of Tinzaparin Thromboprophylaxis in Lung Cancer Patients with High Thromboembolic Risk: A Prospective, Observational, Single-Center Cohort Study"

_cancers, 2024, doi:10.3390/cancers16071442_

Round 1
Reviewer 1 Report
Comments and Suggestions for Authors
1- Specify which kit was used for the measurement of Anti-Xa activity?
-
2- It would be beneficial if the authors could provide clearer definitions of what constitutes intermediate, prophylactic, or therapeutic doses of enoxaparin. Additionally, specifying the first-line treatment received by the cancer patients would enhance clarity and understanding.
Certain structural adjustments in English are necessary to enhance the scientific rigor of the text.
Reviewer 2 Report
Comments and Suggestions for Authors
The authors described a single-center cohort study in which they followed patients with lung cancer treated with Tanzaparin for 6 months.
The topic is interesting; however, data presentation should be improved.
Introduction_ The authors should improve this session by reporting more recent studies on the topic
Results_
• Table 1 and Table 4 could be summarized in the text better explaining their significance for the study.
• Table 3 is the same as Table 2 and the sentences in lines 177-178 are not cited in Table 2
• The description of Figure 1 on lines 203-204-205 cannot be explained in the data presented. The box plot does not represent any significant difference between day 10 and 3 months of treatment in the general population and not between patients who did or did not develop a bleeding event. It is not well explained in the text.
• Section 3.4 could be expanded by analyzing the difference between Anti-Xa levels measured based on the type of treatment.
Reviewer 3 Report
Comments and Suggestions for Authors
I have read the manuscript by Kouvela et al with great interest. Although the study is marked by several limitations, it however brings a novel insight concerning the thromboprophylaxis of lung-cancer patients.
- Why immunotherapy is associated with an increased risk of VTE? Please mention the possible mechanisms.
- There are many confounding factors: given the mean age, the patients did not receive previous anticoagulation for another condition (e.g. atrial fibrillation). Moreover, in the context of cardiovascular risk scores, certain patient had indication for aspirin or other P2Y12 inhibitor.
- The study lacks certain biological baseline characteristics, that could have influenced the outcome (e.g. C-reactive protein, diabetes mellitus with HbA1c etc)
- Please redefine the conclusions of the study: could tinzaparine be a viable alternative cu Enoxaparine? Please bring arguments for both options.
Best regards,
The Reviewer
Comments on the Quality of English LanguageEnglish review by a native speaker
Reviewer 4 Report
Comments and Suggestions for Authors
May you add some pathophysiological remarks as interrelation between hypercoagulopathy and lung cancer? Did you observe a greater incidence of thrombo-embolic complications in small cell lung cancer, often associated to a para-neoplastic syndrome?
Round 2
Reviewer 2 Report
Comments and Suggestions for Authors
The authors have made suggested changes responding to each point in depth.
Reviewer 3 Report
Comments and Suggestions for Authors
Some aspects have been improved accordingly, other observations were rejected by the study protocol (?!)- the authors cannot answer to an observation by invoking the flaws in the study design.